# Pharmacological Probes to Validate Biomarkers for Analgesic Drug Development

**DOI:** 10.3390/ijms23158295

**Published:** 2022-07-27

**Authors:** Johannes van Niel, Petra Bloms-Funke, Ombretta Caspani, Jose Maria Cendros, Luis Garcia-Larrea, Andrea Truini, Irene Tracey, Sonya C. Chapman, Nicolás Marco-Ariño, Iñaki F. Troconiz, Keith Phillips, Nanna Brix Finnerup, André Mouraux, Rolf-Detlef Treede

**Affiliations:** 1Mature Products Develoment, Grünenthal GmbH, 52099 Aachen, Germany; hansvanniel@home.nl; 2Translational Science & Intelligence, Grünenthal GmbH, 52099 Aachen, Germany; petra.blomsfunke@gmail.com; 3Mannheim Center for Translational Neurosciences (MCTN), Department of Neurophysiology, University of Heidelberg, 69120 Mannheim, Germany; ombretta.caspani@medma.uni-heidelberg.de (O.C.); rolf-detlef.treede@medma.uni-heidelberg.de (R.-D.T.); 4Development DMPK, Welab Barcelona, 08028 Barcelona, Spain; jcendros@welab.barcelona; 5Lyon Neurosciences Center Research Unit Inserm U 1028, Pierre Wertheimer Hospital, Hospices Civils de Lyon, Lyon 1 University, 69100 Lyon, France; larrea@univ-lyon1.fr; 6Department of Human Neuroscience, Sapienzia University, 00185 Rome, Italy; andrea.truini@uniroma1.it; 7Wellcome Centre for Integrative Neuroimaging, FMRIB Centre, Nuffield Department of Clinical Neurosciences, University of Oxford, John Radcliffe Hospital, Oxford OX3 9DU, UK; irene.tracey@ndcn.ox.ac.uk; 8Eli Lilly and Company, Arlington Square, Bracknell RG12 1PU, UK; sonya_chapman@lilly.com; 9Department of Pharmaceutical Technology and Chemistry, School of Pharmacy and Nutrition, University of Navarra, 31009 Pamplona, Spain; nmarco@alumni.unav.es (N.M.-A.); itroconiz@unav.es (I.F.T.); 10Eli Lilly and Company, Erl Wood, Bracknell GU20 6PH, UK; phillips_keith_geoffrey@lilly.com; 11Danish Pain Research Center, Department of Clinical Medicine, Aarhus University, 8000 Aarhus, Denmark; finnerup@clin.au.dk; 12Institute of Neuroscience (IoNS), UCLouvain, B-1200 Brussels, Belgium

**Keywords:** biomarkers, analgesic, pain, drug development, PK/PD, proof of mechanism, proof of concept

## Abstract

There is an urgent need for analgesics with improved efficacy, especially in neuropathic and other chronic pain conditions. Unfortunately, in recent decades, many candidate analgesics have failed in clinical phase II or III trials despite promising preclinical results. Translational assessment tools to verify engagement of pharmacological targets and actions on compartments of the nociceptive system are missing in both rodents and humans. Through the Innovative Medicines Initiative of the European Union and EFPIA, a consortium of researchers from academia and the pharmaceutical industry was established to identify and validate a set of functional biomarkers to assess drug-induced effects on nociceptive processing at peripheral, spinal and supraspinal levels using electrophysiological and functional neuroimaging techniques. Here, we report the results of a systematic literature search for pharmacological probes that allow for validation of these biomarkers. Of 26 candidate substances, only 7 met the inclusion criteria: evidence for nociceptive system modulation, tolerability, availability in oral form for human use and absence of active metabolites. Based on pharmacokinetic characteristics, three were selected for a set of crossover studies in rodents and healthy humans. All currently available probes act on more than one compartment of the nociceptive system. Once validated, biomarkers of nociceptive signal processing, combined with a pharmacometric modelling, will enable a more rational approach to selecting dose ranges and verifying target engagement. Combined with advances in classification of chronic pain conditions, these biomarkers are expected to accelerate analgesic drug development.

## 1. Introduction

There is an urgent need for novel analgesics with more favourable safety and tolerability profiles and better efficacy than those currently available, especially in patients with neuropathic and other chronic pain. Over the last two decades, most novel substances with promising antinociceptive properties in preclinical models failed to demonstrate efficacy in clinical trials. The reason for the frequent disparity between the outcomes of preclinical and clinical studies is unclear, and several explanations have been proposed [1,2]. One important reason could be the lack of well-validated biomarkers in the field of pain research. Biomarkers fulfil an array of roles during modern drug development, including demonstration of target engagement, stratification of patients likely to benefit from treatment and prognosis of disease progression [2].

Many drug developments outside the analgesic (and psychopharmaceutic) arena are based on objective, reproducible and quantifiable biomarkers [3] of a target physiological function, for example, microbial growth inhibition, enzyme inhibition, blood pressure and gastric juice production. They can be equally measured in preclinical and clinical pharmacology experiments and can thus be used for translation between the preclinical and clinical situation. Provided that they correlate with the anticipated clinical outcome, they may also be used as surrogate endpoints in clinical trials. Comparable, translatable, broadly applicable, validated biomarkers do not exist for analgesic drug development at this time [4]. In the field of analgesia, the currently recommended outcome measures for clinical trials are patient-reported outcomes (PROs), such as self-reports of pain intensity, daily functioning and quality of life. PROs are intrinsically subjective assessments that reflect how a patient feels and functions. They are thus not only influenced by the beneficial action of an analgesic drug but also by cognitive, affective and social factors, such as anxiety, depression, expectations about treatment efficacy, medication and disease history, marital status and employment status. These non-pharmacological factors generate a large amount of interindividual variability, preventing the use of PROs as pharmacodynamic markers of target engagement. Obviously, PROs cannot be assessed in animals and thus cannot be translated between preclinical and clinical studies. In many failed Phase 3 clinical trials with investigational analgesic drugs, it was not known whether the physiological function targeted by the drug was modulated at the selected drug dose.

Development of pharmacodynamic biomarkers, if they do not already exist, is achieved during preclinical testing. Their first application may be demonstration of target engagement relative to drug concentration in animals. The tools of translational pharmacometry are applied to predict, based on preclinical results, the drug concentrations needed for target engagement in humans. This determines the drug doses that should be shown to be safe and adequately tolerated in Phase 1 and 2 clinical trials. Biomarkers are then used to confirm that a candidate drug engages with its pharmacological target and modulates the target’s physiological function in relation to the drug concentration in humans. If appropriate, biomarkers are also used in Phase 2 studies to identify and select the patient population likely to benefit from the candidate drug.

We postulate that the availability of validated pharmacodynamic biomarkers would improve the success rate of costly Phase 3 analgesic clinical trials as follows:Confirmation of target engagement in preclinical studies. Preclinical evidence of anti-nociceptive efficacy from classical behavioural animal experiments may be strengthened by biomarker assessments.Estimates of the doses that must turn out to be safe and tolerated in early clinical trials and that are needed to achieve target engagement in humans. The possibility of early deselection of drugs with insufficient target engagement in Phase 1 clinical trials conducted in a limited number of healthy volunteers will reduce the failure rates of Phase 3 trials, which may make investments in analgesic development programs more attractive.Identification of patients who are likely to benefit from the candidate drug based on an early evaluation of target engagement, independently of PROs.

In 2016, the Innovative Medicines Initiative (IMI), a public–private partnership between the European Union (EU) and the European Federation of Pharmaceutical Industries and Associations (EFPIA), launched a call to improve the care of patients suffering from acute or chronic pain (H2020-JTI-IMI2-10, topic 3). One of the objectives of this call was the identification and validation of functional biomarkers of pain for future analgesic drug development [5]. Building upon the fact that pain emerges from neuronal activity within the nociceptive system and that analgesics should modulate this neuronal activity in one or more neuronal compartments, a consortium of researchers from academia and industry proposed that such functional biomarkers could be derived from electrophysiological and functional neuroimaging techniques. These techniques are readily available to assess, equally in humans and animals, nociceptor activation thresholds, peripheral nerve excitability, spinal and brainstem processing of nociceptive input and nociception-related cortical activity. However, the methodologies of these measurements are currently neither well standardised nor validated and, possibly for this reason, have seldom played a role in the clinical development of analgesics. These techniques either directly measure neuronal activity or use proxies that are dependent on the functional state of the nociceptive system. Importantly, these measurements of nociceptive function in humans all have the potential for back translation to preclinical species and could thus provide a vital bridge between preclinical and clinical stages of analgesic development.

A prerequisite for validating biomarkers for analgesic drug development is to identify pharmacological probes that can be used to evaluate the sensitivity of the tested biomarkers to drug-induced effects on nociceptive processing. Furthermore, because nociceptive processing takes place at peripheral, spinal and supraspinal levels, pharmacological probes are needed to validate biomarkers at these different levels. In this publication, we describe our methodology to identify drugs that could be used as pharmacological probes for the validation of pharmacodynamic biomarkers of nociceptive processing in humans and discuss the advantages and limitations of the identified probes. Based on these considerations, four randomized clinical studies have been initiated for the validation of biomarkers [6,7,8,9,10,11] (Figure 1).

## 2. Methods

The work described here was performed by the BioPain subtopic of the IMI-PainCare project [13], which is a consortium of researchers from academia and the pharmaceutical industry that combines and mutually shares expertise available to both parties. Its goal is fulfilment of the objectives of Call H2020-JTI-IMI2-10B of the IMI. Details of the studies conducted in healthy subjects to identify and validate pharmacodynamic biomarkers of nociceptive processing in humans are published elsewhere [6,7,8,9,10,11].

### 2.1. Criteria for Selecting Pharmacological Probes

Drugs that could serve as pharmacological probes for the validation of biomarkers of drug-induced effects on nociceptive processing should be known to interact with the peripheral and/or central nervous systems and to exert significant effects on nociceptive processing at peripheral and/or central levels. In order to validate biomarker translation between preclinical species and humans, these pharmacological probes must be licensed for human use and commercially available.

Furthermore, validating biomarkers to be used as pharmacodynamic measures of drug-induced physiological effects on nociceptive processing requires evaluation of whether there is a significant relationship between drug concentration at the site of action and the effect of the drugs on the tested biomarkers. For this reason, drugs used as probes to validate pharmacodynamic biomarkers of nociception should have pharmacokinetic (PK) profiles allowing for assessment of the biomarkers at relevantly different drug concentrations, preferably using repeated measurements performed over a short period, such as a single study day. This requires the pharmacological probes to have a short time to peak concentration (T_max_) and a short elimination half-life (t_½_). Additional desirable PK properties are apparent volume of distribution greater than real physiological volume and low plasma protein binding.

Finally, several practical aspects should be considered, such as ability to administer the pharmacological probe in a non-hospital laboratory setting, pharmacokinetic profiles independent of genetic polymorphism and acceptable tolerability profiles. Several analgesic or antinociceptive drugs have active metabolites. The presence of such metabolites is undesirable, as it would result in an increased variability in drug response. In the not uncommon situation of a divergence between the time profiles of the parent plasma concentration, the metabolite plasma concentration and the pharmacodynamic effect, there would be no certainty as to whether the pharmacodynamic effect is to be ascribed to the parent, the metabolite or both.

For these reasons, drugs were considered as potential pharmacological probes for the validation of pain biomarkers if all of the following applied:Evidence that the drug interacts with the central or peripheral nervous system, preferably with one specific compartment;Drug registered across the EU, preferably as an analgesic, and commercially available in an oral formulation;Sufficient evidence for target engagement and modulation of nociceptive processing in humans. This could be a marketing authorisation as an analgesic, literature data evidencing clinical analgesic efficacy or literature or in-house data indicating a relevant effect of the drug on at least some of the selected biomarkers of nociceptive processing;No active drug metabolites;T_max_ not exceeding 2 h and, preferably, a t_1/2_ of about 2 to 12 h;No dependency of PK parameters on genetic polymorphism (e.g., CYP2D6). This dependency could be accounted for in pharmacometric analysis but would result in a relevantly increased cost and workload and would not have added value for biomarker validation;Tolerability profile indicating that side effects (e.g., sedation or nausea) would not interfere excessively with the biomarker assessments at the tested doses.

Consequently, not eligible were:NSAIDs (including dipyrone/metamizole, aspirin and cyclo-oxygenase-II inhibitors), propacetamol/paracetamol, corticosteroids, sulfasalazine, leflunomide, 5-amino-salicylic acid, resveratrol, triptans, tolterodine and diacerein because they primarily target processes distal to peripheral nociceptive nerve terminals (e.g., inflammatory processes);All local anaesthetics and all fentanyl analogues, buprenorphine, nalbuphine, pethidine, clonidine, dexmedetomidine, ketamine, anaesthetic gases and capsaicin because they are not available in an oral formulation across the EU;Hydrocodone, pentazocine, butorphanol, cebranopadol, levo-alpha-acetyl-methadol (LAAM), cannabinoids, curcumin, caffeine, calcitonin, palmitoylethanolamide, sucrose, substances without an international non-proprietary name (INN), herbal medicines and traditional Chinese medicines because they are not authorised medicines across the EU;Flupirtine and dextropropoxyphene because they were discontinued in the EU market for safety reasons;Retigabine because its worldwide manufacture was discontinued; andControlled-release and abuse-deterrent formulations of known analgesics because formulations must be fast-acting, oral and commercially available.

### 2.2. Search for Candidate Pharmacological Probes

In June 2018, Medline and clinicaltrials.gov were searched to identify candidate drugs with analgesic properties for consideration by IMI-PainCare; subsequently, four clinical trials were planned and registered with EudraCT [6,7,8,9], the designs of three of which have been published [10,11,12]. “Analgesics” [MeSH Major Topic] was the primary search term in Medline; this was further refined with “pain” [MeSH Major Topic], “systematic review” [Publication Type] and “humans” [MeSH Terms], eventually resulting in the inspection of 761 publications (see Figure 2 for a flow diagram). In the clinicaltrials.gov database, “pain” was searched as a disease/condition, and “analgesics” as an intervention/treatment, resulting in the review of 26 drug names. A repeat search of EMA and FDA websites, as well as clinicaltrials.gov, in July 2022 yielded 34 entries in the field of neurology, of which only one was an analgesic medication (oliceridine, a non-morphinan MOR agonist); however, this analgesic is for i.v. application. A simultaneous repeat search in Medline yielded 287 additional publications, but no new possible candidate pharmacological probe emerged.

Summaries of product characteristics of candidate drugs [14] were the preferred sources of information on registration status, PK, metabolites with potentially relevant interaction with the effect of the parent drug and on side effect profiles that might unfavourably interact with biomarker assessments. When needed, additional literature research was performed in Medline. In addition, information available from pharmaceutical companies participating in the project was consulted.

### 2.3. Additional Constraints for Preclinical/Clinical Study Design

Among the candidate medications, our plan was to select three drugs: one as a pharmacological probe preferentially targeting nociception at the peripheral level, one as a probe targeting nociception at the spinal level and one as a probe targeting nociception at the supraspinal level. An additional advantage of similarity of the PK profiles of the pharmacological probes is that serum concentration and pharmacodynamic assessments can be performed by one identical sampling scheme for all three pharmacological probes.

## 3. Results

Figure 3 provides a flow diagram of the drug selection and deselection process. Our search yielded 26 substances (see Table 1). Of these, 13 had marketing authorisation as an analgesic, including four with a specific authorisation for neuropathic pain (pregabalin, gabapentin, duloxetine and amitriptyline) and one with a marketing authorisation for trigeminal neuralgia (carbamazepine). Studies examining oxcarbazepine, lacosamide, lamotrigine, topiramate and valproate showed conflicting results but suggested that these drugs are effective over placebo in subgroups of patients with neuropathic pain [15,16]. For the remaining eight substances, literature or a quotation in clinicaltrials.gov suggested that they were tested for use as an analgesic. However, we did not find any positive results from placebo-controlled, randomised, double-blind trials with these substances.

Of the 13 substances with marketing authorisation as an analgesic, eight are µ-opioid receptor (MOR) agonists, five of which have a morphinan molecular skeleton (codeine, morphine, diacetylmorphine (heroin), oxycodone and hydromorphone). Codeine itself does not have any analgesic activity and requires CYP2D6-mediated in vivo conversion to morphine to exert analgesic activity. Morphine, diacetylmorphine, oxycodone and hydromorphone have relevant active metabolites. Morphinan-type MOR agonists are often used as a control in preclinical and clinical research and are even considered potential gold-standard analgesics for chronic and severe pain. Nevertheless, these substances were not selected as potential probes because the presence of their active metabolites complicates pharmacometric analysis. Of the three non-morphinan MOR agonists, tramadol undergoes CYP2D6-mediated activation to its metabolite with MOR activity, whereas the parent molecule has non-opioid analgesic activity by itself. The two remaining substances, tapentadol and methadone, are available across the EU, are void of active metabolites and have short T_max_ values. Their metabolisms are not (tapentadol) or only slightly (methadone) subject to genetic polymorphism.

Of the 13 substances with marketing authorisation as an analgesic, five substances are not MOR agonists and have been authorised for the treatment of neuropathic pain: pregabalin, gabapentin, carbamazepine, amitriptyline and duloxetine. Metabolites relevantly contributing to the parent drugs’ activity are known for amitriptyline and carbamazepine. The T_max_ values are 4 and 12 h for amitriptyline and carbamazepine, respectively. The T_max_ of duloxetine is 6 h. The eligibility of duloxetine as a pharmacological probe to validate pain biomarkers is further limited by its long t_1/2_ and its CYP2D6-dependent metabolism. Hence, amitriptyline, carbamazepine and duloxetine are not obvious choices for biomarker validation studies, for which a T_max_ not exceeding 2 h is required. Further disadvantages of amitriptyline, carbamazepine and duloxetine are their long t_1/2_-values of 25, 36 and 8–17 h, respectively. Pregabalin and gabapentin have an EU marketing approval for neuropathic pain, are available across the EU, are void of active metabolites, have short T_max_-values (1 and 2–3 h, respectively) and have comparable reported t_1/2_-values of 6.3 and 5–7 h, respectively.

The five substances without marketing authorisation as an analgesic but with a few positive study results for (neuropathic) pain are oxcarbazepine, topiramate, lacosamide, lamotrigine and valproate. In 2015, the Neuropathic Pain Special Interest Group (NeuPSIG) of the International Association for the Study of Pain (IASP, [15]) recommended weakly against valproate. Valproate has an FDA boxed warning against hepatotoxicity and pancreatitis. Hence, valproate is not an obvious choice as a pharmacological probe. Moreover, in 2015, NeuPSIG remained inconclusive in their recommendations about oxcarbazepine, topiramate, lamotrigine and lacosamide. Oxcarbazepine has an active metabolite (10-hydroxycarbazepine; see Figure 4 for time to peak concentration and elimination data) and a T_max_ of 4 h, whereas valproate has a T_max_ of 3–5 h. Hence, oxcarbazepine is not an obvious choice as a pharmacological probe to validate pain biomarkers. Topiramate, lamotrigine and lacosamide have T_max_ values compatible with the requirements for evaluating biomarkers of nociceptive processing over a single study day. Active metabolites are not known for topiramate, lamotrigine and lacosamide. Cochrane reviews concluded that topiramate is without evidence of efficacy in diabetic neuropathic pain [17], that there is no convincing evidence that lamotrigine is effective in treating neuropathic pain [18] and that lacosamide has limited efficacy in the treatment of peripheral diabetic neuropathy [19]. Given some subtype specificity for NaV1.7 and some evidence of clinical efficacy, lacosamide was considered the best available choice.

## 4. Discussion

The choice of medications that can be used to validate functional biomarkers of nociception in humans is limited to drugs that have demonstrated sufficient safety by achieving market authorisation, thus excluding the highly specific probes commonly used in preclinical studies, such as the synthetic opioid peptide DAMGO, NMDA and conotoxins. Furthermore, these probes may be unfit to assess target engagement in specific compartments of the nociceptive system, as most channels and receptors are widely distributed in the nervous system. We therefore categorized pharmacological probes available for human use according to their presumed effect on a given compartment of the nociceptive system. In order to differentiate between the different effect compartments (peripheral nerve, spinal cord or brain), a complex hierarchical statistical analysis including network analysis and identification of latent variables will be necessary; this is scheduled to be completed as part of IMI-PainCare (see Figure 1).

### 4.1. Probes to Validate Biomarkers of Target Engagement at Peripheral Level

For these probes, we evaluated sodium channel blockers (lacosamide, lamotrigine and topiramate). Preclinical data suggest ectopic impulse generator sites as potential effect compartments [24]. Cochrane reviews concluded that topiramate is without evidence of efficacy in diabetic neuropathic pain [17], that there is no convincing evidence that lamotrigine is effective in treating neuropathic pain [18] and that lacosamide has limited efficacy in the treatment of peripheral diabetic neuropathy [19]. It thus appears that the evidence for efficacy is best for lacosamide, and we chose it as the pharmacological probe to validate pain biomarkers at the peripheral level. However, lacosamide is used as an antiepileptic and is thus also active at the cortical level. Lacosamide is thought to preferentially modulate nociceptive processing at the peripheral level through an effect on the slow inactivation of voltage-gated sodium channels [25]. Lacosamide is also thought to modulate collapsin response mediator protein 2-mediated neurotrophic signals [25]. In the EU, lacosamide has marketing authorisation for epilepsy but not for neuropathic pain. However, its clinical effect on neuropathic pain has been studied over 10 to 18 weeks in patients with painful diabetic neuropathy [26,27,28,29]. In two out of three placebo-controlled studies, superiority over placebo was reported for the 200 mg twice-daily regimen, although at the cost of a 12.2% rate of premature discontinuations. Despite this rate of discontinuations related to adverse events, no safety issue in particular of lacosamide became apparent from these studies. Additional support for the analgesic efficacy of lacosamide in neuropathic pain conditions comes from two further clinical studies. One showed the effect of lacosamide in patients with Na_v1_._7_ mutation-related small fibre neuropathy [30]; the other found that in painful small-fibre neuropathy, lacosamide normalised the firing pattern of C fibres, reduced heat and pain thresholds and reverted nociceptor excitability in iPSC (human induced pluripotent stem cell)-derived nociceptors, suggesting a specific effect of lacosamide on the function of peripheral nociceptors [31]. Based on these six studies, we concluded that there is convincing evidence that lacosamide has a relevant effect on nociceptive processing in the peripheral compartment. Absorption of lacosamide after a single oral dose is fast—in the range of 30 min. Active metabolites are not known for lacosamide. Additional data on lacosamide relevant for pharmacometric evaluation are provided in Table 2, and a simulated PK profile is provided in Figure 4. The only disadvantage of lacosamide is its slow elimination. The selection of lacosamide is thus a compromise. Its relatively slow elimination could be a challenge for the assessment of concentration effect relationships over a single study day. The authors of several papers have proposed that blockers of sodium channels specific to nociceptive afferents may be the way forward [32,33], but such subtype-specific blockers have proven difficult to develop, possibly due to the small extracellular domains of these channels.

### 4.2. Probes to Validate Biomarkers of Nociceptive Processing at the Spinal Level

We evaluated gabapentinoids (gabapentin, pregabalin), for which there is good preclinical evidence of antinociceptive effects at the spinal level [39] and early clinical evidence that its administration may modulate experimentally induced secondary hyperalgesia in healthy subjects [40]. However, absorption of gabapentin is slow, saturable and thus not dose-linear and not very predictable, whereas the absorption of pregabalin absorption is fast and predictable. This renders pregabalin the preferred option to validate biomarkers of nociceptive processing at the spinal level. According to its antiepileptic activity, pregabalin also acts at the cortical level. Pregabalin binds to the α_2_-δ subunit of presynaptic, voltage-dependent calcium channels that are widely distributed throughout the central and the peripheral nervous system [41,42,43,44]. It thereby reduces the release of several neurotransmitters, such as glutamate, noradrenaline, serotonin, dopamine and substance P [45,46,47,48,49,50], ultimately modulating synaptic transmission at the spinal level [51]. Additional data on pregabalin relevant for pharmacometric evaluation are provided in Table 2, and a simulated PK profile is provided in Figure 4. This justifies the selection of pregabalin as a probe for the spinal level.

### 4.3. Probes to Validate Biomarkers of Nociceptive Processing at the Cortical Level

As probes for efficacy at cortical level, we evaluated two opioids: tapentadol and methadone. Brain areas involved in nociceptive processing have a high expression of mu opioid receptors (MOR) [52,53,54]. For methadone, a very long t_1/2_ of 19–55 h is reported, as compared to tapentadol (4 h). The safety profile of methadone is also unfavourable compared to tapentadol, particularly for studies with healthy subjects. Therefore, we chose tapentadol as a pharmacological probe for biomarkers of nociceptive processing at the supraspinal level. A spinal site of action is also possible because of high opioid receptor expression in the superficial dorsal horn. A dual mode of action is suggested for tapentadol: agonism at the MOR and inhibition of the synaptosomal reuptake of noradrenaline [55,56]. Additional data on tapentadol relevant for pharmacometric evaluation are provided in Table 2, and a simulated PK profile is provided in Figure 4.

In conclusion, lacosamide, pregabalin and tapentadol are our preferred probes for the identification and validation of biomarkers of nociceptive processing at peripheral, spinal and supraspinal levels, respectively. All three substances can safely be assumed to interact with their target compartments. However, none of these three drugs act specifically in these target compartments. All three substances may be expected to interact, at least to some extent, with nociceptive processing at the supraspinal level: tapentadol through its MOR-agonism and the other two because of their known antiepileptic activity. Nevertheless, lacosamide, pregabalin and tapentadol may be expected to differ in the relative size of their effects on nociceptive processing at the peripheral, spinal and supraspinal levels, leading to differential effects on the biomarkers derived from neuronal activity at these levels. Moreover, lacosamide, pregabalin and tapentadol differ in their modes of action [25,45,46,47,48,49,50,51,52,56].

### 4.4. Proposed Pharmacometric Analyses

Because all available analgesic medications act on more than one compartment of the nociceptive system, some advanced pharmacometric techniques must be added to evaluate the differential effects of the proposed model probes on the pharmacodynamic biomarkers of the nociceptive system. These include differences in drug concentration between compartments (time and concentration curves after single oral doses, Figure 4) and regression analyses across biomarkers to model contributions of several compartments to the efficacy of each probe (Figure 1).

Finally, evaluating the pharmacodynamic properties of the biomarkers requires assessment of drug effects on the biomarkers at relevantly different drug concentrations. The concentration at the site of action is a crucial determinant of the magnitude of a drug effect, and this concentration at the target site may be significantly delayed as compared to drug plasma concentration. For example, in humans, the T_max_ of pregabalin in the cerebrospinal fluid (CSF) occurs about 8 h post dose, which indicates delayed distribution to the brain [57]. For lacosamide, the T_max_ in plasma and CSF in humans occurs at approximately the same time [34]. Corresponding data for tapentadol have not been published. To account for this possible delay, assessments of drug effects at time points after peak plasma concentration are needed. Drugs with short T_max_ and t_1/2_ values are desirable because they allow for assessment of drug-induced effects at relevantly different drug concentrations using repeated measurements performed over a single study day.

The restrictive criteria used to select the optimal pharmacological probes for biomarker validation (i.e., the absence of active metabolites, absence of genetic polymorphism of drug metabolism, oral administration in a non-hospital setting and rapid pharmacokinetics) do not limit the applicability of the biomarkers for the development of future analgesic drugs. Once effective biomarkers have been identified and properly validated, it will be feasible to design clinical pharmacology studies with drug candidates that would not satisfy these selection criteria. For example, the biomarkers could be used to assess target engagement of drugs administered non-orally. Clinical pharmacology studies extending over several days may be performed at dedicated institutions. Drugs with metabolism not susceptible to genetic polymorphism may be evaluated in genetically profiled subjects. Studies may be designed to account for the presence of more than one active substance.

## 5. Conclusions

The availability of pharmacodynamic biomarkers of nociceptive processing will optimise the role of pharmacometrics in analgesic drug development to (i) describe the quantitative relationship between drug exposure and drug-induced changes in nociceptive processing, (ii) extrapolate these relationships, as found in preclinical research, to corresponding relationships in healthy subjects and (iii) establish a quantitative translational framework to predict the concentrations required in humans for effective target engagement and modulation [58]. This innovative approach will add to ongoing efforts aimed at an improved understanding of and controlling for variance in placebo effect sizes in analgesic drug trials [1,15].

We postulate that biomarkers of nociceptive processing will enable the selection of a dose range in Phase 1 studies that is adequate to verify the safety and tolerability of drug concentrations needed for target engagement in humans. Next, it will allow for the verification of target engagement in Phase 1 or Phase 2 studies. In later stages, model-informed drug development can integrate these relationships to select doses for clinical testing in patients. Notwithstanding the above points, the eventual and final proof of clinical efficacy of a candidate analgesic drug will still have to come from the demonstration of relevant effects on clinical endpoints in confirmatory Phase 3 clinical trials. However, biomarkers could permit patient stratification and enrichment in Phase 3 clinical trials, as encouraged by the EMA/CHMP/970057/2011 Guideline.

## Figures and Tables

**Figure 1 ijms-23-08295-f001:**
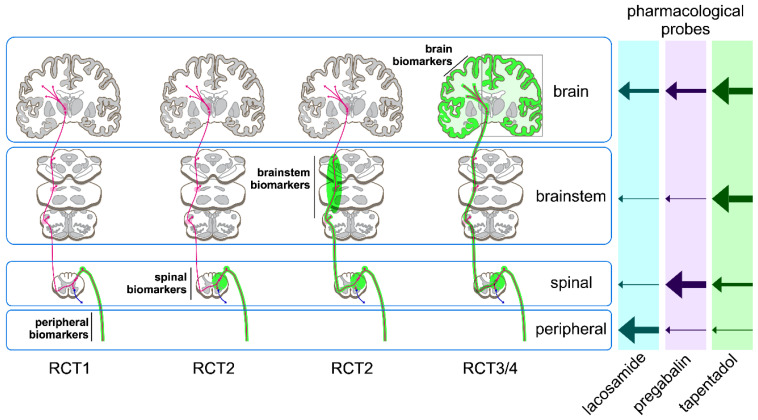
The aim of the BioPain project of the IMI-PainCare consortium is to validate a set of pharmacodynamic biomarkers of nociceptive processing derived from non-invasive measures of nociceptive processing at the peripheral, spinal, brainstem and brain levels. Whereas some biomarkers are selective readouts for a given compartment of the nociceptive system (e.g., small-fibre perception threshold tracking as a readout of nociceptive processing at the level of the peripheral nervous system), other biomarkers are dependent on the state of nociceptive processing along the entire neuraxis (e.g., laser-evoked brain potentials that are sequentially processed and transmitted at peripheral, spinal cord and brain levels). These biomarkers will be tested across four parallel clinical studies (RCT1 [6,11], RCT2 [7,12], RCT3 [8,10] and RCT4 [9]) using three pharmacological probes (lacosamide, pregabalin and tapentadol) that are expected to predominantly affect nociceptive processing at peripheral, spinal and brain levels, respectively. Nonetheless, all three probes are active in multiple compartments. Hence, complex hierarchical modelling and estimation of latent variables will be used in addition to PK-PD modelling.

**Figure 2 ijms-23-08295-f002:**
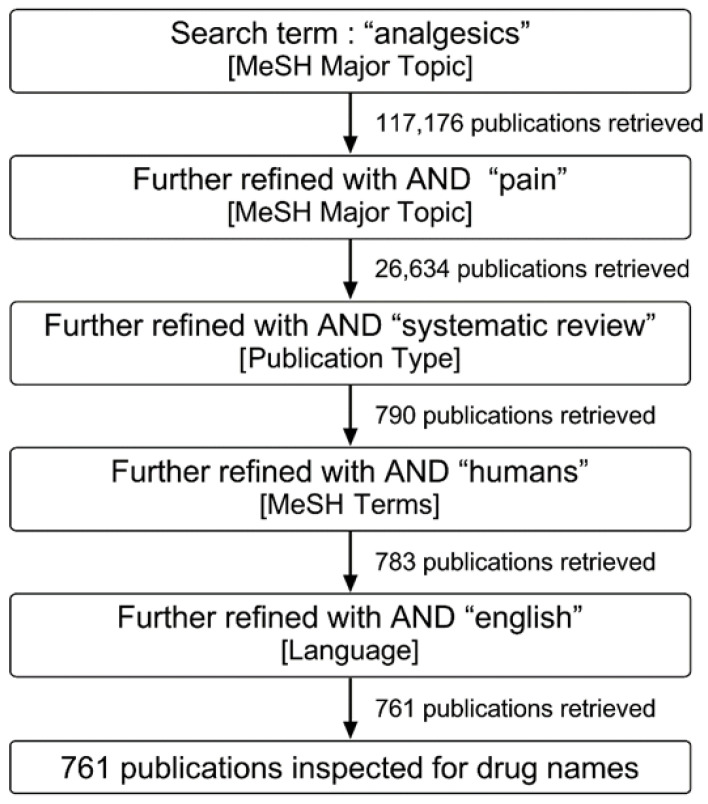
Flow diagram of Medline search.

**Figure 3 ijms-23-08295-f003:**
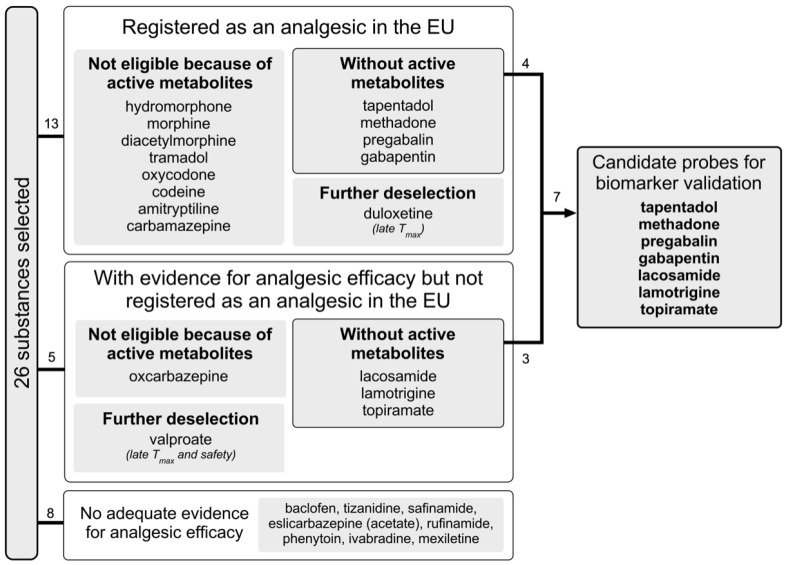
Flow diagram of selection process.

**Figure 4 ijms-23-08295-f004:**
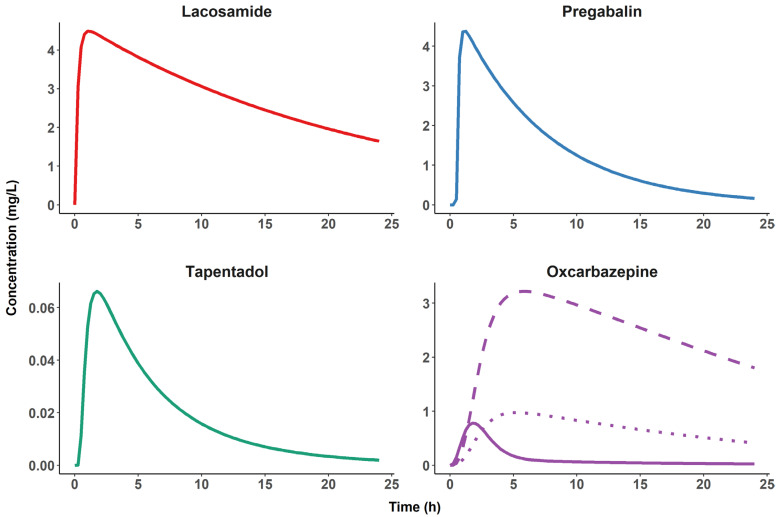
Typical plasma pharmacokinetic profiles of four candidate pharmacological probes. Simulated plasma concentrations for a single dose, oral administration of 200 mg of lacosamide [20], 150 mg of pregabalin [21], 100 mg of tapentadol [22] and 300 mg of oxcarbazepine (including metabolites) [23] during a 24 h study period. Note the fast elimination of the oxcarbazepine parent compound (solid line) compared to the other pharmacological probes and the presence of two active metabolites, S-(+)-10-hydroxycarbazepine (dashed line) and R-(−)-10-hydroxycarbazepine (dotted line), making oxcarbazepine an unsuitable candidate for the planned experimental designs.

**Table 1 ijms-23-08295-t001:** Main characteristics of candidate drugs for validation of biomarkers of nociception ^1^.

Substance	Evidence for Use in Pain	Active Metabolites? ^10^	T_max_ (h) ^4^	t_1/2_ (h) ^3^	Polymor-phism? ^9^	Mode of Action ^8^	Active in Compartment		
							P ^2^	S ^2^	B ^2I^
**Substances with marketing authorisation as an analgesic and without relevant active metabolites**									
Tapentadol	Registered as an analgesic	No	1.25	4	no	M, NA		+	+
Pregabalin	Registered for neuropathic pain	No	1	6.3	no	CC, NT		+	+
Methadone	Registered as an analgesic	Not reported	1.5–3	19–55	minor, CYP2D6, CYP2B6	M		+	+
Gabapentin	Registered for neuropathic pain	No	2–3	5–7	no	CC, NT		+	+
Duloxetine	Registered for neuropathic pain	No	6	8–17	CYP2D6	SNRI		+	+
**Substances with marketing authorisation as an analgesic and with relevant active metabolites**									
Hydromorphone	Registered as an analgesic	Hydromorphone-3-glucuronide	0.5–1	2–3	no	M		+	+
Morphine	Registered as an analgesic	Morphine-6-glucuronide	1	2	no	M		+	+
Diacetyl-morphine ^5^	Registered as an analgesic	6-acetyl-morphine, morphine	NA ^5^	0.03–0.05	no	M		+	+
Tramadol	Registered as an analgesic	(+)-O-demethyl-tramadol	1–2	5–6	CYP2D6	M, NA, S		+	+
Oxycodone	Registered as an analgesic	Oxymorphone, noroxycodone	1–1.5	3	CYP2D6	M		+	+
Codeine ^6^	Registered as an analgesic	Morphine	-	-	CYP2D6				
Amitriptyline	Registered for neuropathic pain	Nortriptyline	4	25	CYP2D6	SNRI	+	+	+
Carbamazepine	Registered for trigeminal neuralgia	Yes	12	36	no	SC	+	+	+
**Substances with evidence of analgesic activity but without marketing authorisation as an analgesic**									
Lacosamide	yes	No	0.5–4	12–13	no	SC	+	+	+
Valproate	yes	Not reported	3–5	14	no	G, SC, HDI			+
Topiramate	yes	Not clinically relevant	1.4–4.3	18–22	no	SC, GA	+		+
Lamotrigine	No positive studies	No	2.5	33	no	SC	+		+
Oxcarbazepine	One positive study	10-hydroxy-carbazepine	4.5	1–3	no	SC	+	+	+
**Substances without substantial evidence of analgesic activity**									
Baclofen	no	Not reported	0.5–1.5	3–4	no	GB, CC		+	
Tizanidine	no	Not reported	1	2–4	no	α	+	+	+
Safinamide	no	No	2–3	20–30	no	MB		+	+
Eslicarbazepine	no	Not clinically relevant	2.5–3	9–11	no	SC	+		+
Rufinamide	no	No	4–6	6–10	no	SC	+		+
Phenytoin	no	Not known	? ^7^	7–42	no	SC			+
Ivabradine	no	Yes	1	2–11	no	I*_f_*	+		+
Mexiletine	no	Yes	3.0	9–11	yes	SC, AA	+		+

^1^ Data are taken from SmPCs or literature [13]. ^2^ P = peripheral compartment, S = spinal compartment, B = supraspinal compartment. ^3^ t_1/2_ = elimination half-life. ^4^ T_max_ = time needed to reach peak plasma concentration. ^5^ Oral administration of diamorphine (diacetylmorphine, i.e., heroin) results in measurable blood concentrations of morphine but not diamorphine or 6-acetylmorphine. The amount of circulating morphine provided by an oral dose of diamorphine was only 79% of that available from an equal amount of morphine. Hence, T_max_ is not reported. ^6^ Codeine itself has no analgesic activity and requires CYP2D6-dependent conversion to morphine to become active. In so-called poor metabolisers (i.e., subjects without functional CYP2D6), codeine has no analgesic efficacy. Hence, data on target compartment, time to peak plasma concentration and t_1/2_ of codeine are not regarded as informative. ^7^ Absorption of phenytoin is described as slow and variable. Time to peak plasma concentration is not given. ^8^ α = α_2_-receptor agonist; AA = antiarrhythmic; CC = calcium channel binding; G = acting on γ-aminobutyric acid (GABA) levels in the brain; GA = interacting with GABA-A receptors; GB = GABA_B_ receptor activation; HDI = histone deacetylase inhibition; I_f_ = acting on the I_f_ ion current, which is a mixed Na+–K+ inward current; M = µ-opioid receptor agonism; MB = monoamine oxidase B inhibition; NA = noradrenaline reuptake inhibition; NT = neurotransmitter release; SC = interaction with sodium channels; SNRI = serotonin and noradrenaline reuptake inhibition. ^9^ Provided information indicates whether elimination is subject to genetic polymorphism. ^10^ Current regulatory guidelines require extensive research to identify metabolites and assess their pharmacological activity, if any. “No” means that absence of active metabolites can be assumed; “Not reported” means that the drug was developed before current guidelines were in place to ensure the absence of active metabolites.

**Table 2 ijms-23-08295-t002:** Physicochemical, biopharmaceutical and pharmacokinetic properties of the selected pharmacological probes.

Property	Lacosamide	Pregabalin	Tapentadol
MW (g/Mol)	250.30	159.23	221.34
Solubility (g/L)	0.465 ^a^	>30 ^b^	1.16 ^c^
Lipophilicity (Log P)	0.728 ^d^	−1.35	2.87
pKa	>12 ^e^	4.2//10.6	9.6//10.28
BCS class	I	I	I
Bioavailability (%)	≈100	>90%	32%
Fu	>0.85	1	≈0.8
CL (L/h)	1.92 ^#^	4.02–4.85	91.8
V (L)	42 ^#^	39.2 ^#^	540
Unaltered fraction in urine	0.4	1	0.03
Metabolism	CYP2C9, CYP2C19 and CYP3A4 Relative contribution of each CYP still unknown	-	70% conjugation 13% CYP2C9 and CYP2C19 2% CYP2D6
CNS data	Concentration ratio: CSF/Serum ^†^ 0.85 ^f,^, 0.641 ^g^ Brain/Plasma * 0.553 ^h^	Concentration ratio: CSF/Plasma * ≈0.1 ^i^	Concentration ratio: Brain/Plasma * ≈4 ^j^

^#^ For a 70 kg body weight; ^†^ human data, * rat data. Abbreviations: MW, molecular weight; BCS class, Biopharmaceutical Classification System; Fu, fraction unbound; Cl, clearance; V, volume of distribution; CNS, central nervous system; CSF, cerebral spinal fluid. Source: Summary of Product Characteristics and drug label. Otherwise: ^a^ Predicted using ALOGPS, Virtual Computational Chemistry Laboratory, 2005; ^b^ Chemistry review, FDA Center for Drug Evaluation and Research, application number: 22–488; ^c^ predicted using Estimation Program Interface (EPI) Suite, US EPA; ^d^ ACD/Labs; ^e^ Chemistry review, FDA Center for Drug Evaluation and Research, application number: 22–255; ^f^ May et al. (2015) [34]; ^g^ Michelhaugh et al. 2015 [35]; ^h^ Koo et al. (2011) [36]; ^i^ Feng et al. (2001) [37]; ^j^ Schröder et al. (2011) [38].

## Data Availability

All quoted data are available in the public domain.

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
