# Peer review of "Pharmacological Probes to Validate Biomarkers for Analgesic Drug Development"

_ijms, 2022, doi:10.3390/ijms23158295_

Round 1
Reviewer 1 Report
Summary:
This review is well written and covers a very topical problem, since we are currently dealing with an opioid crisis, especially in the USA, where it is a leading cause of death with a record number of > 100,000 cases reported last year. Analgesic drug development for better treatment approaches is urgently needed with current developments often failing to translate from pre-clinical to clinical efficacy and/or tolerability. This publication describes the approach for the identification of EU approved drugs that might be suitable as pharmacological probes to be used for the urgently needed standardization and validation of biomarkers within the nociceptive system in humans.
Recommendation: Minor revision
Main concern:
Line 196: The literature search for candidate pharmacological probes was performed over 4 years ago, which is quite a long time for the pharmaceutical market. Even if not much has happened in terms of EU approved analgesics that would fit the inclusion citeria, a more recent search would be more appropriate.
Results/Figure 2: Opioids, or more specifically molecules with morphinan structure are despide their safety concerns still the gold standard analgesic for chronic and severe pain. It is often used in pre-clinical and clinical trials as a control. It seems also one of the best validated analgesics in terms of efficacy. It seems these morphin-derivatives would be a very good candidate for a pharmacological probe to validate biomarkers and should not be as easily dismissed because it does not fit all criteria. A strong argument against using morphin-derivatives as probes should be presented. While active metabolites might cause variability, their PK can also be determined to correct for some of these variabilities. It would be appropriate to give some additional arguments against the use of morphin-derivatives as a probe in the text since it seems like an appropriate positive control drug despite presence of well known active metabolites. Almost more importantly, it should also be noted that only because active metabolites of the other drugs are not reported, does not mean they do not exist.
Line 309: The selected sodium channel blockers are not specific for sodium channel subtypes. The different sodium channel isoforms are widely distributed throughout the body in neuronal and non-neuronal tissue (e.g. skeletal muscle, heart muscle, central nervous and peripheral nervous tissue including nociceptive and non-nociceptive neurons). This should be discussed.
Line 314: The reasons for selecting lacosamide are not clear based on the arguments presented in the discussion. Please give reasons why exactly the evidence for efficacy is best for lacosamide.
Line 359: Pregabalin clearly has central nervous activity as antiepileptic drug and targets the same receptors in the brain and periphery that is is thought to target in spinal level to induce analgesia. Why then was it selected as a spinal level specific probe? Only because it does affect the spinal level neurons does not mean that this is the main mode of action for its analgesia. Synaptic transmission is modulated on all levels, not just the spinal level, making the statement in line 365 inaccurate. Please correct and discuss more.
Line 369ff: Since mu-opioid receptors are also expressed in periphery and spinal level where they are clearly involved in nociceptive signalling, please discuss the analgesic effects tapentadol has on the other levels as well. Why was it selected as a cortical level specific probe? Please discuss.
Generally, the crosstalk of the selected probes between the 3 systems/compartments will make biomarker validation more complex and should not be considered to be compartment specific. While this is mentioned in the discussion, it still sounds like the 3 drugs/probes may act antinociceptive preferentially through the compartment they were assigned to in this report. All three drugs act on all 3 compartments, because all of them cross the blood-brain barrier and the blood-spinal cord barrier and have their relevant targets in all these compartments. This should be better clarified. The benefits of compartment specific probes should also be more clear, and how effects on the other compartments will interfere with the biomarker validation. It seems difficult to determine the drug concentration at the “site of action” when there are multiple different sites of action in tissues with different drug concentrations after a single administration. It seems appropriate to measure and validate the biomarkers in all 3 compartments for all three selected probes, only then can one conclude which compartment contributes most to the analgesic effects.
Minor comments:
Line 200: reference error for Figure 1
Lines 219, 220: again Figure and Table reference errors. Please fix throughout the document.
Table 1 should be redesigned since it is not very well readable. Some shading or colors, bold lines where appropriate, bold text where appropriate, e.g. in title row, more care about the line breaks, etc.; and again no active metabolites is not the same as none reported!
Line 328ff: Please specify which of these studies involve human subjects.
Variability of pain relieve in placebo groups in human clinical trials should briefly be discussed.
Author Response
Reply to Reviewer #1
This review is well written and covers a very topical problem, since we are currently dealing with an opioid crisis, especially in the USA, where it is a leading cause of death with a record number of > 100,000 cases reported last year. Analgesic drug development for better treatment approaches is urgently needed with current developments often failing to translate from pre-clinical to clinical efficacy and/or tolerability. This publication describes the approach for the identification of EU approved drugs that might be suitable as pharmacological probes to be used for the urgently needed standardization and validation of biomarkers within the nociceptive system in humans.
Recommendation: Minor revision
Reply: We appreciate this positive evaluation
Main concern:
- Line 196: The literature search for candidate pharmacological probes was performed over 4 years ago, which is quite a long time for the pharmaceutical market. Even if not much has happened in terms of EU approved analgesics that would fit the inclusion criteria, a more recent search would be more appropriate.
Reply: A repeat search of EMA and FDA websites as well as clinicaltrials.gov in July 2022 yielded 34 entries in the field of neurology, of which only one was an analgesic medication (Oliceridine, a non-morphinan µ-receptor agonist). This analgesic is for i.v. application and hence no new possible candidate pharmacological probe emerged during the first four years of IMI-PainCare activities. The manuscript has been updated accordingly (see page 8 lines 7-12).
- Results/Figure 2: Opioids, or more specifically molecules with morphinan structure are despite their safety concerns still the gold standard analgesic for chronic and severe pain. It is often used in pre-clinical and clinical trials as a control. It seems also one of the best validated analgesics in terms of efficacy. It seems these morphine-derivatives would be a very good candidate for a pharmacological probe to validate biomarkers and should not be as easily dismissed because it does not fit all criteria. A strong argument against using morphine-derivatives as probes should be presented. While active metabolites might cause variability, their PK can also be determined to correct for some of these variabilities. It would be appropriate to give some additional arguments against the use of morphine-derivatives as a probe in the text since it seems like an appropriate positive control drug despite presence of well known active metabolites. Almost more importantly, it should also be noted that only because active metabolites of the other drugs are not reported, does not mean they do not exist.
Reply: We now discuss both PROs and CONs of morphinan-type opiates as positive control candidates (page 9 lines 12-16). We agree with the reviewer that only because active metabolites of the other drugs are not reported, does not mean they do not exist. However, for new substances, regulators require that all possible metabolites are identified already in the preclinical testing and confirmed later on in vivo in humans and that all potentially relevant metabolites are tested for pharmacodynamic activity by sensitive receptor binding and receptor activation studies in vitro and by testing in vivo. There is very low probability that a drug approved for marketing since 2000 has an unknown active metabolite. This is quite different for older drugs developed in times when modern in vitro receptor binding and activation tests were not available.
- Line 309: The selected sodium channel blockers are not specific for sodium channel subtypes. The different sodium channel isoforms are widely distributed throughout the body in neuronal and non-neuronal tissue (e.g. skeletal muscle, heart muscle, central nervous and peripheral nervous tissue including nociceptive and non-nociceptive neurons). This should be discussed.
Reply: We agree with the Reviewer, and the revised manuscript now expands on this issue (page 12 lines 7-10). In fact, there is no good subtype-specific set of sodium channel blockers and we mention some potential reasons (small extracellular domains, Emery et al. 2016, Bennett et al. 2019).
- Line 314: The reasons for selecting lacosamide are not clear based on the arguments presented in the discussion. Please give reasons why exactly the evidence for efficacy is best for lacosamide.
Reply: We selected lacosamide because (1) it has some subtype specificity for NaV1.7, (2) it has mixed evidence for clinical efficacy in diabetic neuropathy, which makes it a candidate for more sophisticated studies, and (3) the PK profile (including the absence of active metabolites) of lacosamide is better suitable for our project than any alternative candidate drugs. This has been clarified in the revised manuscript, page 10 lines 18-23.
- Line 359: Pregabalin clearly has central nervous activity as antiepileptic drug and targets the same receptors in the brain and periphery that is is thought to target in spinal level to induce analgesia. Why then was it selected as a spinal level specific probe? Only because it does affect the spinal level neurons does not mean that this is the main mode of action for its analgesia. Synaptic transmission is modulated on all levels, not just the spinal level, making the statement in line 365 inaccurate. Please correct and discuss more.
Reply: We appreciate this comment and have added a new Figure to illustrate the effect compartments that are (1) assessed by the proposed biomarkers, and (2) modulated by the proposed drugs. A network analysis approach with complex hierarchical modelling and estimation of latent variables is clearly appropriate and will actually be used in the IMI-PainCare project. This is now explained (page 11 lines 1-6). Ideally, one would have selected drugs specific for each of the 3 compartments. However, such drugs do not exist and we had to choose drugs that interact with all 3 compartments, though to a different extent.
- Line 369ff: Since mu-opioid receptors are also expressed in periphery and spinal level where they are clearly involved in nociceptive signaling, please discuss the analgesic effects tapentadol has on the other levels as well. Why was it selected as a cortical level specific probe? Please discuss.
Reply: We appreciate this comment. As explained in our reply to the preceding comment, we have added a Figure that illustrates the effect compartments assessed by the proposed biomarkers, and modulated by the proposed drugs.
- Generally, the crosstalk of the selected probes between the 3 systems/compartments will make biomarker validation more complex and should not be considered to be compartment specific. While this is mentioned in the discussion, it still sounds like the 3 drugs/probes may act antinociceptive preferentially through the compartment they were assigned to in this report. All three drugs act on all 3 compartments, because all of them cross the blood-brain barrier and the blood-spinal cord barrier and have their relevant targets in all these compartments. This should be better clarified. The benefits of compartment specific probes should also be more clear, and how effects on the other compartments will interfere with the biomarker validation. It seems difficult to determine the drug concentration at the “site of action” when there are multiple different sites of action in tissues with different drug concentrations after a single administration. It seems appropriate to measure and validate the biomarkers in all 3 compartments for all three selected probes, only then can one conclude which compartment contributes most to the analgesic effects.
Reply: We appreciate this comment and agree that it is appropriate to measure and validate the biomarkers in all three compartments for all three selected probes. This will be achieved by a joint analysis of the data collected in all four RCTs, using a network analysis approach with complex hierarchical modelling and estimation of latent variables. This is illustrated in the new Figure, and better explained , as it is now explained page 5 lines 20-21, page 6 line 27, and page 11 lines 1-6.
Minor comments:
- Line 200: reference error for Figure 1
Reply: Thanks. Corrected in revision.
- Lines 219, 220: again Figure and Table reference errors. Please fix throughout the document.
Reply: Thanks. Corrected in revision.
- Table 1 should be redesigned since it is not very well readable. Some shading or colors, bold lines where appropriate, bold text where appropriate, e.g. in title row, more care about the line breaks, etc.; and again no active metabolites is not the same as none reported!
Reply: We improved the table format as suggested. We rephrased as “none reported”
- Line 328ff: Please specify which of these studies involve human subjects.
Reply: We now specify that both studies were done in humans and none in rodents (page 11 line 28).
- Variability of pain relieve in placebo groups in human clinical trials should briefly be discussed.
Reply: We now comment on effect sizes for placebo, page 14 lines 27-29.
Reviewer 2 Report
This paper presents a study conducted to highlight the need for identification and validation of functional Pharmacodynamic biomarkers of pain for analgesic drug development. It also emphasized the need for development of methodologies to measure nociceptive function in humans for better analgesic development.
The author has done a good job in highlighting the role of functional biomarkers as well as provide information to help identify drugs which have the potential to be used as pharmacological probes for validation of pharmacodynamic biomarkers.
However, detailed explanation needs to be provided regarding the methodology used to identify and validate pharmacodynamic biomarkers. Also, manuscript should also provide information about the work currently being done in the area of functional biomarkers.
Results and discussion part of the manuscript seems well researched but introduction and the need for this study should be highlighted more so as to provide complete information regarding pharmacodynamic biomarkers.
Author Response
Reply to Reviewer #2
This paper presents a study conducted to highlight the need for identification and validation of functional Pharmacodynamic biomarkers of pain for analgesic drug development. It also emphasized the need for development of methodologies to measure nociceptive function in humans for better analgesic development.
The author has done a good job in highlighting the role of functional biomarkers as well as provide information to help identify drugs which have the potential to be used as pharmacological probes for validation of pharmacodynamic biomarkers.
Reply: We appreciate this positive evaluation
However, detailed explanation needs to be provided regarding the methodology used to identify and validate pharmacodynamic biomarkers. Also, manuscript should also provide information about the work currently being done in the area of functional biomarkers.
Reply: We appreciate this comment and have added a Figure that illustrates the effect compartments that are assessed by the proposed biomarkers. We also quote some recent studies that have employed the proposed concepts, page 5 lines 20-21, and page 8 lines 1-2.
Results and discussion part of the manuscript seems well researched but introduction and the need for this study should be highlighted more so as to provide complete information regarding pharmacodynamic biomarkers.
Reply: To better highlight the need and potential for the study, a new Figure has been included to explain the proposed biomarker validation approach. P. 5 line 20-21, Fig. 1

Round 2
Reviewer 2 Report
Authors have addressed all my commentsI. I recommend this review for publication.